# Continuous Glucose Monitoring Improves Weight Loss and Hypoglycemic Symptoms in a Non-Diabetic Bariatric Patient 14 Years After RYGB: A Case Report

**DOI:** 10.3390/reports8040200

**Published:** 2025-10-08

**Authors:** Carolina Pape-Köhler, Christine Stier, Stylianos Kopanos, Joachim Feldkamp

**Affiliations:** 1Klinikum Bielefeld—Rosenhöhe, Bariatric Clinic, An der Rosenhöhe 27, 33647 Bielefeld, Germany; carolina.pape-koehler@klinikumbielefeld.de; 2Department of Surgery, Interdisciplinary Endoscopy, University Medicine Mannheim, Heidelberg University, Theodor-Kutzer-Ufer 1-3, 68167 Mannheim, Germany; christine.stier@umm.de; 3Department of General Internal Medicine, Endocrinology and Diabetes, Infectiology, Klinikum Bielefeld-Mitte, Medical School and University Medical Center OWL, Bielefeld University, Universitätsstraße 25, 33615 Bielefeld, Germany; joachim.feldkamp@klinikumbielefeld.de

**Keywords:** continuous glucose monitoring, post-bariatric hypoglycemia, glucose variability, dumping syndrome, patient empowerment

## Abstract

**Background and Clinical Significance**: Roux-en-Y gastric bypass (RYGB) significantly alters glucose metabolism, yet managing glucose variability in patients undergoing bariatric surgery remains challenging. Continuous Glucose Monitoring (CGM) offers real-time insights into glucose fluctuations and may support long-term metabolic management in this population. This case highlights the utility of CGM in identifying postprandial glycemic variability and guiding dietary adjustments. **Case Presentation**: A 45-year-old female, 14 years post-RYGB, presented with symptoms including postprandial fatigue, nocturnal cravings, and unexplained weight gain, despite adherence to nutritional guidelines. Her BMI had decreased from 52 kg/m^2^ (pre-surgery) to 29 kg/m^2^. She was provided with a CGM device (FreeStyle Libre 3) by the clinical team and instructed to monitor glucose without modifying her routine initially. Data revealed significant glycemic variability, with peaks up to 220 mg/dL and hypoglycemic dips to 45 mg/dL. Based on this, she adjusted her diet by reducing non-complex carbohydrates and increasing vegetables, proteins, and complex carbohydrates. Within two weeks, her symptoms improved, including better sleep and energy levels, accompanied by a 3 kg weight loss following dietary adjustments informed by CGM feedback. **Conclusions**: This case suggests how CGM can empower patients having undergone bariatric surgery to manage glucose fluctuations through informed self-management. The patient’s ability to identify and address glucose variability without formal intervention highlights CGM’s potential as a supportive tool in long-term care. While further studies are needed, this case suggests CGM may benefit similar patients experiencing postprandial symptoms or weight regain after bariatric surgery.

## 1. Introduction and Clinical Significance

Bariatric surgery, and, in particular, Roux-en-Y gastric bypass (RYGB), is among the most effective interventions for sustained weight loss and remission of obesity-related comorbidities. Beyond weight control, RYGB profoundly alters glucose metabolism through mechanisms involving accelerated nutrient delivery to the jejunum, altered incretin secretion, and changes in insulin sensitivity [1,2]. While these effects are beneficial in reducing type 2 diabetes risk, a significant proportion of post-RYGB individuals experience glycemic disturbances, most notably postprandial hypoglycemia (also referred to as late dumping syndrome) [3]. These episodes may present with nonspecific symptoms such as fatigue, nausea, neuroglycopenia, or weight fluctuations, and can occur several years after surgery [4].

Continuous glucose monitoring (CGM) has emerged as a valuable tool for identifying glycemic variability in people with diabetes, and its application is increasingly being explored in post-bariatric populations [5,6]. CGM offers advantages over provocation tests, as it provides continuous, real-world data on glucose fluctuations during daily living, including nocturnal periods that would otherwise go undetected [7,8]. Studies have demonstrated that CGM may uncover a higher frequency of hypoglycemic episodes compared to mixed-meal tolerance tests in RYGB patients [9]. However, despite its diagnostic potential, CGM remains underutilized in the post-bariatric setting, and there is limited literature on its use as a facilitative tool for patient self-management rather than purely as an investigational device [10].

What is exceptional about this case is that it documents a person without diabetes, more than a decade after RYGB, who was able to independently identify and modify dietary triggers using CGM feedback. This approach allowed her to stabilize glycemic variability, improve symptoms, and achieve meaningful weight loss without formal dietary counseling or pharmacological intervention. To our knowledge, few reports have highlighted the empowering role of CGM in post-bariatric individuals as an adjunct for long-term follow-up and self-directed care. This makes the present case noteworthy and of relevance to clinicians managing patients with persistent symptoms years after bariatric surgery.

This case report was prepared in accordance with the CARE guidelines for case reports [11]. All relevant items from the CARE Checklist have been addressed in the manuscript and the Checklist is provided as a for review.

The CGM device (FreeStyle Libre 3, Abbott, Austin, TX, USA) was purchased through routine clinical care at our institution. The manufacturer had no role in device provision, data collection, analysis, or manuscript preparation. None of the authors have any financial or professional affiliations with Abbott.

## 2. Case Presentation

We present a case of a 45-year-old female who underwent Roux-en-Y gastric bypass (RYGB) 14 years prior, with a history of class III obesity. Prior to RYGB at age 31, her maximum weight was 142 kg (BMI 52 kg/m^2^). She had attempted multiple weight loss strategies including supervised diet programs, pharmacological therapy (e.g., orlistat for 6 months), and structured physical activity, all without sustained benefit. Following RYGB, her lowest post-surgical weight was 78 kg (BMI 29 kg/m^2^), which she maintained for several years. Her medical history included hypertension (controlled without medication after surgery), and mild iron-deficiency anemia in remission. She had no history of diabetes, liver disease, or other endocrine disorders. Family history was negative for diabetes or early cardiovascular disease. She reported no alcohol abuse, smoking, or recreational drug use. The patient had consistently participated in long-term follow-up care and had adhered to a regimen of nutritional supplements, with no abnormalities noted in her laboratory results.

However, she reported experiencing latent symptoms, including postprandial fatigue accompanied by mild nausea. Additionally, she described nocturnal cravings and morning fatigue, alongside persistent generalized tiredness and reduced energy levels. The patient also noted a slight, unexplained weight gain despite no changes in her dietary habits.

First-line evaluation for common post-bariatric causes of fatigue and nausea was performed. This included complete blood count, electrolytes, renal and liver function, thyroid function (TSH), iron studies (ferritin, iron), vitamin B12 and folate, and 25-OH vitamin D (reported above), all without clinically relevant abnormalities. There was no diabetes (fasting glucose 92 mg/dL, HbA1c 5.3%), and the patient was taking no agents affecting glucose metabolism. In view of symptoms occurring several hours after meals and occasional nocturnal cravings, post-bariatric hypoglycemia (PBH)/late dumping was considered. Contemporary guidance emphasizes that provocation tests such as OGTT can precipitate severe dumping and are not recommended for PBH evaluation; MMTT may be used selectively but dynamic tests are not required for diagnosis, and recent consensus statements advise a physiology-based approach with documented low glucose during typical daily life.

A Continuous Glucose Monitoring (CGM) system (FreeStyle Libre 3^TM^, Abbott) was provided to the patient, with instructions to maintain her usual dietary and lifestyle habits (shown in Figure 1). No further interventions were implemented at this stage.

Accordingly, rather than perform OGTT/MMTT first, we elected a pragmatic, short-term home CGM to (i) capture 24 h patterns (including nocturnal episodes) under usual diet, (ii) correlate symptoms with glucose trends, and (iii) support immediate dietary titration. We applied a FreeStyle Libre 3 sensor in clinic (posterior upper arm), instructed the patient to scan at least every 8 h to avoid data gaps, and reviewed data at follow-up. This approach aligns with recent guidance that while CGM is not a diagnostic test for dumping/PBH, it can be valuable for monitoring everyday glycemic responses and informing diet and self-management in this setting.

On days one and two of monitoring, the patient’s continuous glucose monitoring (CGM) data revealed significant glycemic variability, characterized by pronounced glucose excursions (shown in Figure 2a,b). Notably, glucose peaks reached as high as 220 mg/dL, while hypoglycemic episodes occurred up to four times per night, with blood glucose levels dropping as low as 45 mg/dL. The glucose profiles demonstrated rapid fluctuations, with a swift increase followed by a similarly rapid decline in serum glucose concentrations. These findings suggest considerable instability in the patient’s glucose regulation.

On the third day, the patient initiated modifications to her dietary habits in response to her observations from the previous two days (shown in Figure 3). No formal intervention was implemented, and the patient did not undergo any specific dietary training, nor was she prescribed a particular diet (e.g., low carbohydrate). Instead, she employed a trial-and-error approach, using the real-time glucose data from the CGM to change her dietary choices.

In particular, she reduced the intake of non-complex carbohydrates that caused rapid glucose spikes and increased the consumption of vegetables, proteins, and complex carbohydrates. This dietary adjustment was associated with an improvement in her well-being. Notably, these changes resulted in a significant reduction in the frequency of nocturnal hypoglycemic episodes, from four occurrences per night to just one. This suggests that the patient’s glycemic control improved primarily through dietary self-regulation, with CGM acting as a facilitator by providing real-time feedback.

Quantitative CGM analysis was performed using FreeStyle Libre 3 data, comparing the initial 48 h (baseline: Days 1–2, before dietary changes) with the final 48 h (follow-up: Days 13–14, after two weeks of monitoring and dietary adaptation). This device is factory-calibrated, requires no user calibration, and has a mean absolute relative difference (MARD) of approximately 9.2%, as per manufacturer specifications. A summary of baseline and follow-up CGM metrics—including mean glucose, time-in-range (TIR), time-below-range (TBR), coefficient of variation (CV), and MAGE—is presented in Table 1. These values illustrate improved glycemic stability by day 14 of monitoring.

Baseline laboratory values revealed a fasting glucose of 92 mg/dL, HbA1c of 5.3%, fasting insulin of 8.1 µU/mL, and C-peptide of 1.4 ng/mL. The lipid panel showed total cholesterol of 176 mg/dL, HDL 61 mg/dL, LDL 96 mg/dL, and triglycerides 88 mg/dL. Micronutrient levels included vitamin B12 at 512 pg/mL, folate 8.4 ng/mL, ferritin 46 ng/mL, iron 87 µg/dL, and 25-OH vitamin D at 31 ng/mL. No major abnormalities were observed post-intervention. The patient had a history of hypertension (currently controlled without medication) and mild iron-deficiency anemia in remission. She was taking a daily multivitamin, sublingual vitamin B12, and vitamin D but no medications affecting glucose metabolism. A pre-intervention dietary recall revealed frequent intake of refined carbohydrates and sugary snacks, particularly in the evening. Meals were often low in protein and fiber. This pattern was consistent with reactive hypoglycemia symptoms and may have contributed to her reported fatigue and weight gain.

By Days 13–14 (after two weeks of monitoring), the patient had established a consistent routine and gained a more comprehensive understanding of the factors influencing the stabilization of her blood glucose levels (shown in Figure 4a,b). The lifestyle modifications she implemented contributed to improved glycemic control. She no longer experienced postprandial nausea, and her sleep quality had significantly improved. She reported waking up feeling rested, without fatigue. Additionally, she experienced a weight loss of 3 kg over the two-week period, further reflecting the positive impact of these changes on her overall health.

A chronological overview of the patient’s clinical course, including surgery, follow-up, onset of symptoms, CGM initiation, dietary adaptation, and outcomes, is summarized in Figure 5.

## 3. Discussion

We report a case commonly encountered in follow-up care: patients presenting with nonspecific symptoms, mild fatigue and subtle weight gain. These patients often struggle to explain their condition, yet they remain dissatisfied and report a general sense of discomfort. Many patients also perceive a reduction in their overall performance capacity and have the feeling that they have failed.

Often, there is a tendency to question the compliance of these individuals, prompting the recommendation to keep a food and symptom diary. They are referred to nutritional counseling, and a follow-up appointment is scheduled for a few weeks later to reassess their progress. This approach, however, rarely resolves the underlying issues. Patients tend to become increasingly frustrated, and their symptoms do not significantly improve. Although these symptoms are sometimes blamed on poor compliance or dietary mistakes, they are more often rooted in altered post-surgical physiology. In our patient, CGM feedback helped her identify and correct dietary triggers linked to glucose variability.

The root cause is more likely related to the altered incretin response induced by the surgical procedure due to duodenal exclusion. The underlying pathomechanism can be explained by the altered anatomy due to the operation.

PBH typically occurs 2–4 h after eating years after RYGB and may present with non-specific symptoms that are difficult to reproduce in clinic. Current guidance cautions against using OGTT in this population because it can provoke severe dumping and is not physiologic; some centers use MMTT selectively, but 2024 consensus guidance does not require dynamic provocation tests for diagnosis [12]. CGM is not recommended as a stand-alone diagnostic test for dumping/PBH; however, it provides continuous, real-world glucose profiles, detects nocturnal/asymptomatic hypoglycemia, and can effectively guide dietary modification—all of which were central to our patient’s care. Prior studies also show CGM uncovers more hypoglycemic episodes than MMTT in post-RYGB cohorts. In our case, CGM served as an adjunctive monitoring/education tool, enabling the patient to identify high-glycemic triggers and rapidly stabilize variability through diet, consistent with these recommendations [13,14].

In gastric bypass surgery, a small stomach pouch with a capacity of approximately 25–40 mL is created, and a jejunal loop, known as the alimentary loop, is anastomosed to this pouch. This loop ranges from 50 to 150 cm in length. The remaining stomach is preserved but no longer participates in the normal gastrointestinal passage, nor does the duodenum, a condition referred to as duodenal exclusion (shown in Figure 6). Digestive secretions are directed into the alimentary loop via a biliopancreatic loop, which also measures 50 to 150 cm. From this point onward, the essential processes of chyme breakdown and complex carbohydrate digestion occurs. The pylorus is excluded from the gastrointestinal tract, and the rate of gastric emptying is primarily determined by the width of the anastomosis. Initially, food does not contact the duodenum, preventing the release of duodenal incretins such as GIP. As a result, food reaches the terminal ileum more rapidly, leading to a sharp rise in blood glucose levels, which causes initial hyperglycemia. The resulting elevated insulin secretion causes a rapid decrease in blood glucose levels. Since this insulin response lags and remains active longer than the elevated glucose levels, hypoglycemia may subsequently occur.

These pathophysiological mechanisms help explain our patient’s symptoms and why she was able to manage them relatively easily. By monitoring her blood glucose levels, she identified that short-chain carbohydrates caused the most significant glucose spikes, followed by hypoglycemia. By altering her diet to include more complex carbohydrates and combining them with proteins, she slowed the carbohydrate passage and absorption time, resulting in more gradual blood glucose rises and a more appropriate insulin response.

Our patient experienced improvements in both quality of life (QoL) and weight without the need for dietary counseling or a strict regimen. She recognized that her well-being was closely tied to glucose variability, and her symptoms were the primary factors influencing her self-management decisions. No formal intervention was necessary, and even two months later, both her glucose variability and weight remained stable.

The patient presented with non-specific but troubling symptoms—fatigue, nocturnal cravings, and weight regain—which are frequently seen in long-term follow-up. Continuous Glucose Monitoring (CGM) revealed pronounced postprandial glycemic variability, including both hyper- and hypoglycemia. Guided by these data, the patient independently modified her diet, resulting in symptom resolution and weight loss. The primary driver of improved glycemic control was the patient’s nutritional changes, while CGM served as a supportive tool that enhanced self-awareness and autonomy. Instead of feeling helpless or passive regarding her health, the patient was able to take responsibility for her well-being and to actively manage her health problem. Similarly to reports by Lupoli et al. [1] and Kefurt et al. [4], we observed that CGM can uncover hypoglycemic events not captured by mixed-meal tests. Our findings are therefore in agreement with existing literature that supports CGM as a useful adjunctive tool in post-bariatric populations [15].

The FreeStyle Libre^TM^ system, being user-friendly and intuitive, is ready for immediate use. A survey conducted during the COVID-19 pandemic demonstrated its effectiveness in managing diabetes in patients during confinement [5,16]. Information obtained through CGM self-monitoring serves as a valuable psychoeducational tool, combining medical guidance with self-care strategies. This approach enables individuals to make more informed decisions about their eating habits, thereby improving follow-up management, either for weight control, symptom management, or metabolic regulation [17,18].

While this case suggests potential benefit from CGM-guided dietary adaptation, the short 14-day observation period limits conclusions regarding long-term efficacy. To avoid overinterpretation, we emphasize that this report reflects an individual patient experience and does not establish causality. However, follow-up at 3 months indicated sustained improvements: the patient reported a total weight loss of 5.8 kg since CGM initiation and continued symptom resolution. These findings support the hypothesis that CGM may serve as a useful adjunct in long-term post-bariatric care, though larger prospective studies are warranted.

Self-management facilitated by CGM has been shown to reduce the time required for therapy adjustments and enhance patient independence, as evidenced by several studies. In diabetic patients, CGM use has resulted in more stable glucose levels and a reduction in HbA1c. These findings suggest that CGM could be beneficial for the long-term follow-up of bariatric patients, promoting the stabilization of glucose levels.

The major strength of this report is the clear demonstration of how CGM feedback can facilitate rapid and effective self-management without formal dietary counseling or pharmacological intervention. The case also documents sustained improvement at three months, strengthening the clinical relevance of the observation. However, as a single case report, the findings lack external validity and cannot be generalized. The short monitoring period (14 days) further limits conclusions regarding long-term efficacy, and dietary changes act as a confounding variable, making causality impossible to establish.

Future research should explore the role of CGM in larger post-bariatric cohorts, ideally with prospective studies assessing both diagnostic accuracy and patient-reported outcomes. Trials comparing CGM-guided dietary self-management with standard dietary counseling would help clarify whether CGM adds value beyond existing follow-up strategies. Finally, integration of CGM into multidisciplinary bariatric follow-up programs could be tested as a means of improving quality of life, detecting PBH earlier, and potentially preventing weight regain.

## 4. Conclusions

Our case presents a common problem in bariatric surgery patients in long-term follow up. We present a case where CGM functioned as a supportive, self-empowering tool that enabled the patient to recognize patterns and adjust her diet accordingly. The improvements observed should be attributed to dietary changes, with CGM acting as a facilitator rather than a direct therapeutic agent.

The findings from a single patient’s experience are not sufficient to draw definitive conclusions. However, these results suggest that CGM may play a valuable role in the long-term follow-up of bariatric surgery patients. CGM could also provide potential benefits for post-bariatric patients who experience weight regain.

Further investigation is needed to confirm our findings in a broader patient cohort. This would present a great, simple, and modern approach to using CGM in the follow-up care of bariatric patients.

## Figures and Tables

**Figure 1 reports-08-00200-f001:**
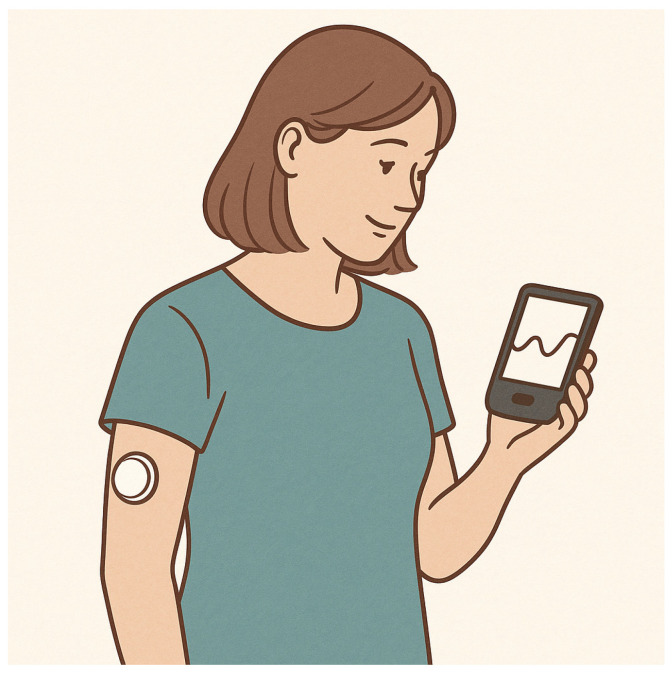
FreeStyle Libre sensor for continuous interstitial glucose monitoring.

**Figure 2 reports-08-00200-f002:**
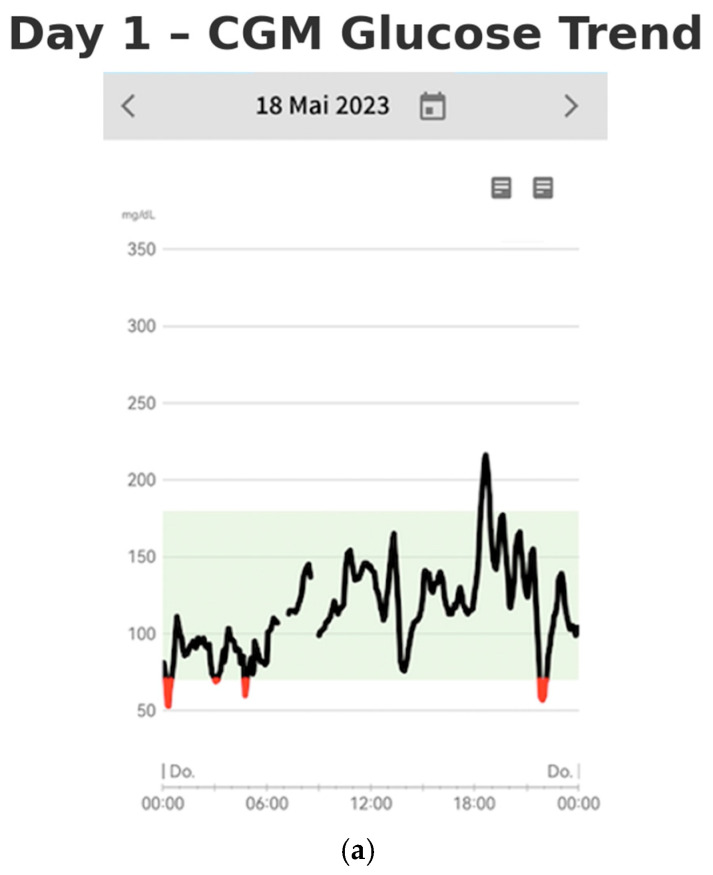
(**a**,**b**) CGM glucose trend on Day 1 and Day 2, showing early postprandial hypoglycemia with variable readings and increased hypoglycemia frequency.

**Figure 3 reports-08-00200-f003:**
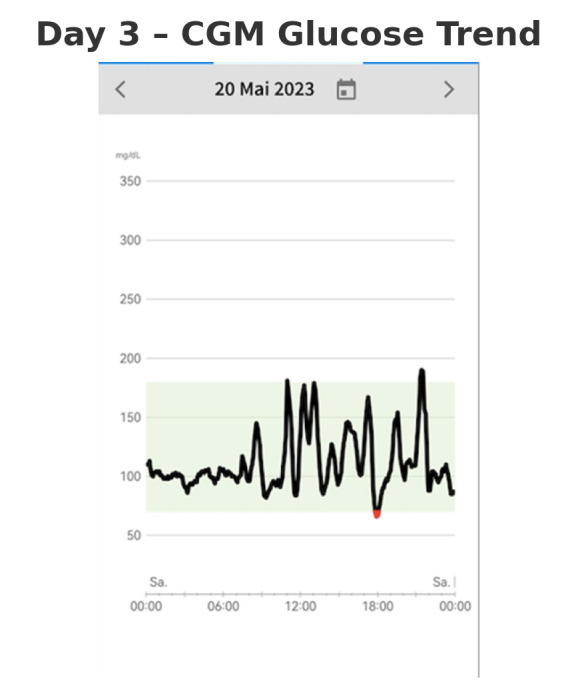
CGM glucose trend on Day 3, revealing highly fluctuating postprandial responses.

**Figure 4 reports-08-00200-f004:**
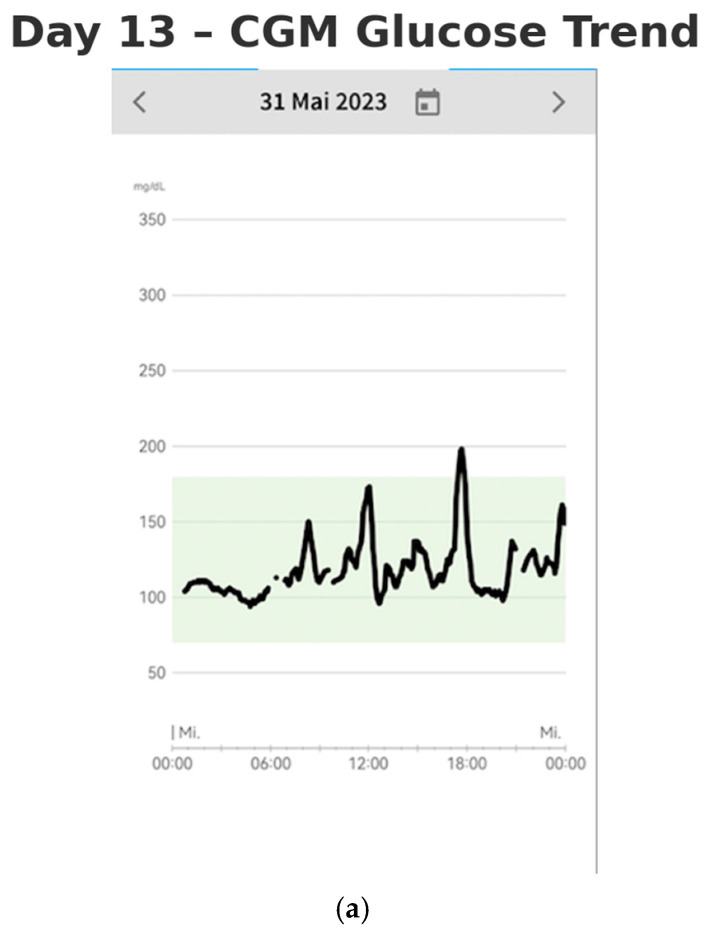
CGM glucose trend on (**a**) Day 13 and (**b**) Day 14, showing improved glycemic stability with reduced glycemic variability.

**Figure 5 reports-08-00200-f005:**
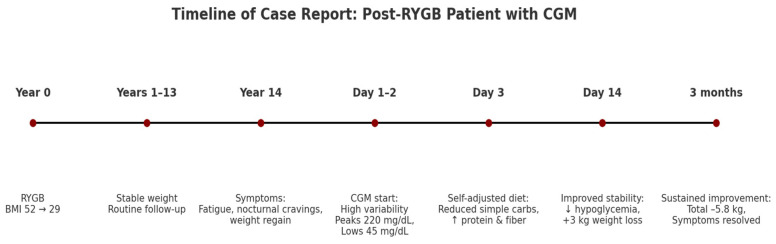
Timeline of case report. Clinical course of a person 14 years after Roux-en-Y gastric bypass (RYGB). After onset of fatigue and weight regain, continuous glucose monitoring (CGM) was initiated, leading to identification of glycemic variability and subsequent patient-driven dietary changes, with symptomatic improvement and sustained weight reduction at 3 months.

**Figure 6 reports-08-00200-f006:**
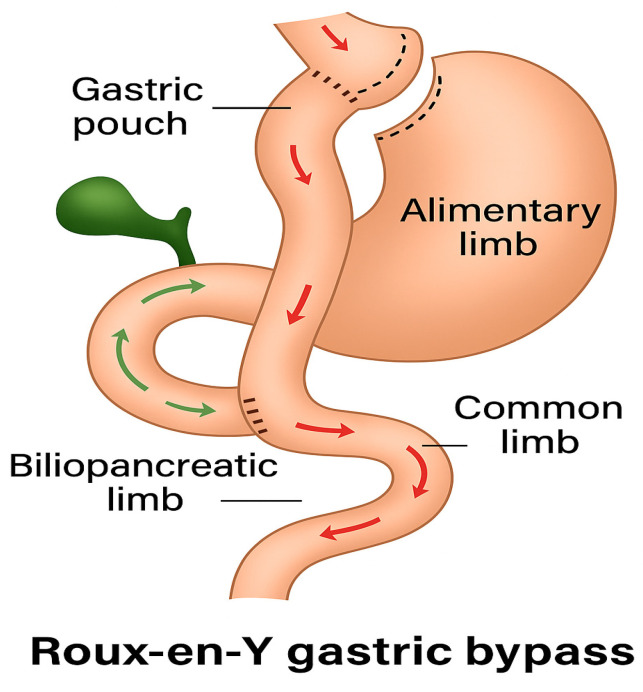
Roux-Y-Bypass: gastric pouch, alimentary limb (red arrows), biliopancreatic limb (green arrows), common limb (red and green arrows).

**Table 1 reports-08-00200-t001:** Quantitative table of CGM evidence.

Metric	Baseline (Days 1–3)	Follow-Up (Days 13–14)
Mean glucose (mg/dL)	134	112
% Time in range (70–140 mg/dL)	64%	82%
% Time below range (<70 mg/dL)	9%	2%
Coefficient of variation (%)	38	28
MAGE (mg/dL)	85	58

## Data Availability

The original contributions presented in this study are included in the article material. Further inquiries can be directed to the corresponding author.

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
