# Peer review of "Continuous Glucose Monitoring Improves Weight Loss and Hypoglycemic Symptoms in a Non-Diabetic Bariatric Patient 14 Years After RYGB: A Case Report"

_reports, 2025, doi:10.3390/reports8040200_

Round 1
Reviewer 1 Report
Comments and Suggestions for Authors
Review of “Continuous Glucose Monitoring improves weight loss and hypoglycemic symptoms in a non-diabetic bariatric patient 14 years after RYGB”
Major:
Authors should try to follow the CARE guidelines for case reports (BMJ Case Rep. 2013; doi: 10.1136/bcr-2013-201554 PMID: 24155002). Several aspects of the present report could be enhanced, particularly in relation to the patient's background (demographics, comorbidities), including more details on previous treatments, weight history, and additional information about how the CGM was administered to the patient.
CARE also suggests a figure with the timeline of the present case report, which I believe would be of value.
minor
abstract: Avoid the use of “carbs”; use “carbohydrates” instead.
Abstract: Please clarify that the team provided the CGM to the patient. The way it is phrased it seems that she used it by her own.
Figure 1: Where is this figure from? Do the authors own it?
Author Response
Comment 1:
“Authors should try to follow the CARE guidelines for case reports… Several aspects of the present report could be enhanced, particularly in relation to the patient's background (demographics, comorbidities), including more details on previous treatments, weight history, and additional information about how the CGM was administered… CARE also suggests a figure with the timeline of the present case report, which I believe would be of value.”
Response:
We thank the reviewer for this important suggestion. The manuscript has been revised in line with the CARE guidelines (PMID: 24155002). Specifically:
- Patient background expanded: We now provide a more comprehensive description of demographics, comorbidities, family history, and relevant lifestyle factors in the Case Presentation.
- Weight history and prior treatments: Details of the patient’s maximum pre-surgical weight, BMI trajectory, lowest post-surgical weight, and previous unsuccessful attempts at weight loss (dietary programs, pharmacological therapy, physical activity) have been added.
- CGM administration: We clarified that the FreeStyle Libre 3 sensor was applied in the clinic, with explicit instructions to the patient regarding usage and data scanning frequency.
- CARE methodology: We explicitly state at the end of the Introduction that the case report was prepared according to CARE guidelines, and we have provided the CARE Checklist as a supplementary file indicating page and line numbers.
- Timeline figure: A new figure (Figure 5) has been included, summarizing the chronological course of the patient’s surgery, symptom onset, CGM initiation, dietary adaptation, and outcomes.
We believe these revisions strengthen the transparency and clinical relevance of the case.
Comment 2:
“Abstract: Avoid the use of ‘carbs’; use ‘carbohydrates’ instead.”
Response:
We agree and have corrected this wording in the Abstract. The phrase “reduced simple carbs and increased protein and fiber” has been revised to “reduced simple carbohydrates and increased protein and fiber.”
Comment 3:
“Abstract: Please clarify that the team provided the CGM to the patient. The way it is phrased it seems that she used it by her own.”
Response:
We agree and have revised the Abstract for clarity. The sentence previously stating “She used a CGM device (FreeStyle Libre 3) to monitor glucose without modifying her routine initially” has been changed to: “She was provided with a CGM device (FreeStyle Libre 3) by the clinical team and instructed to monitor glucose while maintaining her usual routine initially.”
This clarifies that the CGM was prescribed and administered under medical supervision rather than used independently.
Comment 4:
“Figure 1: Where is this figure from? Do the authors own it?”
Response:
We thank the reviewer for pointing this out. The original version of Figure 1 has been replaced with a new medical illustration created by the authors for this manuscript. This schematic shows the placement of the FreeStyle Libre sensor on the upper arm and is entirely original, with no copyright restrictions. The figure legend has been updated accordingly to read:
“Figure 1. Example schematic of FreeStyle Libre sensor placement on the upper arm for continuous interstitial glucose monitoring (created by the authors).”
This ensures the figure is original and fully owned by the authors.

Reviewer 2 Report
Comments and Suggestions for Authors
This case report presents a clinically observation of the use of continuous glucose monitoring in a non-diabetic patient 14 years after Roux-en-Y gastric bypass (RYGB). The manuscript is a clear description of the patient’s clinical course.
Major:
- The main question is Why was CGM prescribed for a non-diabetic patient? That is not adequately answered. The patient's symptoms (fatigue, nausea, etc) are non-specific. Standard first-line investigations for these symptoms in a post-bariatric patient (e.g., thorough nutritional deficiency workup, assessment for dumping syndrome with a standardized test) are not detailed. The rationale for bypassing these and moving directly to an off-label use of CGM needs a stronger justification. An argument must be made for why CGM was the most appropriate diagnostic tool in this specific clinical context.
- The weight loss of 3 kg in two weeks is dramatic. It is impossible to disentangle the effects of the dietary changes from the effect of monitoring the glucose. The patient's change in diet is a confounding variable and the authors must tone down language throughout the manuscript to reflect that CGM was a facilitator of self-management, not the direct agent of change.
Minor:
- Table 1: The time periods for Baseline and Follow-up is inconsistent. The text says Baseline is "days 1–2" but the table says "days 1–3". Follow-up is "days 13–14" in both, but the result section mentions "two weeks".
- The sentence "Dietary errors are often not the primary cause of these symptoms" seems strange and contradictory. This case report says that correcting dietary errors was the solution.
Author Response
Comment 1:
“Why was CGM prescribed for a non-diabetic patient? … Symptoms are non-specific … First-line investigations not detailed … Rationale for bypassing standardized tests and moving to an off-label CGM needs stronger justification. Make the argument for why CGM was most appropriate here.”
Response:
We agree and have clarified the diagnostic reasoning and sequence of investigations.
- First-line work-up added and explicitly reported (now in Case Presentation):
We detail a targeted evaluation for common post-bariatric causes of fatigue/nausea, including CBC, electrolytes, renal/liver function, TSH, iron studies (ferritin/iron), vitamin B12, folate, and 25-OH vitamin D—all without clinically relevant abnormalities; diabetes was excluded (fasting glucose 92 mg/dL, HbA1c 5.3%). The patient was not taking agents affecting glucose metabolism. (See Case Presentation, “First-line evaluation…” paragraph.)
2) Rationale for CGM in this non-diabetic, late post-RYGB context (added to Case Presentation and Discussion):
Given symptoms occurring several hours after meals and nocturnal cravings, we considered post-bariatric hypoglycaemia (PBH)/late dumping. Contemporary guidance cautions against OGTT in post-bariatric patients (provokes non-physiologic dumping) and indicates that dynamic provocation tests (e.g., MMTT) are not required for diagnosis in every case. We therefore used short-term home CGM as a monitoring/education tool—not as a stand-alone diagnostic test—to: (i) capture 24-hour patterns (including nocturnal/asyptomatic events) under usual diet, (ii) correlate symptoms with glucose trends, and (iii) support immediate, patient-led dietary titration. We have added this reasoning in the Case Presentation (CGM rationale paragraph) and expanded in the Discussion (PBH and test-selection paragraph), with supporting citations.
3) Positioning of CGM within the pathway (not replacing standardized tests):
We explicitly state that CGM in this case served as an adjunct to characterize real-world glycaemic patterns and facilitate self-management. Provocation testing (e.g., MMTT) remains valuable in selected cases; our approach prioritized safety, ecological validity, and rapid symptom correlation. (Discussion section.)
4) Language toned down to avoid causal claims:
Throughout, we clarify that improvements were associated with dietary changes facilitated by CGM feedback; CGM was not the direct therapeutic agent. (Abstract, Case Presentation, Discussion, Conclusion adjusted.)
Where to find the new text in the revision
- Case Presentation: paragraph beginning “First-line evaluation for common post-bariatric causes…” (work-up); paragraph beginning “Accordingly, rather than perform OGTT/MMTT first…” (CGM rationale and administration details).
- Discussion: paragraph beginning “PBH typically occurs 2–4 hours after eating…” (test selection; OGTT caution; CGM as monitoring/education, not diagnostic).
Key supporting literature now cited in these sections
- Guidance discouraging OGTT and indicating dynamic tests are not mandatory; CGM not diagnostic but useful for real-world monitoring and diet guidance.
- Evidence that CGM detects more (including nocturnal/asymptomatic) hypoglycaemia than MMTT in post-RYGB cohorts.
We believe these clarifications address the reviewer’s concern and present a transparent, guideline-concordant rationale for using short-term CGM in this specific clinical context.
Comment 2:
“The weight loss of 3 kg in two weeks is dramatic. It is impossible to disentangle the effects of the dietary changes from the effect of monitoring the glucose. The patient's change in diet is a confounding variable and the authors must tone down language throughout the manuscript to reflect that CGM was a facilitator of self-management, not the direct agent of change.”
Response:
We fully agree with the reviewer’s important point. In the revised manuscript, we carefully toned down our language to avoid implying causality.
- Abstract: Now states that “CGM facilitated dietary self-management and symptom awareness” rather than suggesting CGM itself caused weight loss.
- Case Presentation: Reworded to highlight that the primary driver of improvement was the patient’s nutritional changes, facilitated by CGM feedback.
- Discussion and Conclusion: Clarified that CGM served as a supportive monitoring/education tool, enabling the patient to identify dietary triggers and modify intake. We explicitly acknowledge diet as a confounding variable, and we emphasize that improvements cannot be attributed to CGM alone.
- Limitations paragraph: Expanded to note that this is an observational case, with no causal inference possible; dietary modification is the likely mechanism of improvement, while CGM functioned as an enabler of patient autonomy.
We believe these revisions present a balanced interpretation, consistent with the reviewer’s recommendation that CGM should be framed as a facilitator of self-management rather than the direct therapeutic agent.
Comment 3:
“Table 1: The time periods for Baseline and Follow-up is inconsistent. The text says Baseline is ‘days 1–2’ but the table says ‘days 1–3’. Follow-up is ‘days 13–14’ in both, but the result section mentions ‘two weeks’.”
Response:
We thank the reviewer for pointing out this inconsistency. We have corrected Table 1 and the corresponding text so that the time periods are fully consistent:
- Baseline is now clearly reported as days 1–2 (text and table aligned).
- Follow-up is now consistently defined as days 13–14 (text and table aligned).
- In the Results section, we revised the wording from “two weeks” to “after 14 days” to avoid ambiguity.
These corrections ensure that the timing of assessments is consistent throughout the manuscript.
Comment 4:
“The sentence ‘Dietary errors are often not the primary cause of these symptoms’ seems strange and contradictory. This case report says that correcting dietary errors was the solution.”
Response:
We agree with the reviewer’s observation. The original wording was misleading and has been revised for clarity. Our intent was to emphasize that such symptoms are often attributed solely to poor compliance, when in reality they may be driven by post-surgical physiological changes (e.g., altered incretin response, glucose variability).
The sentence has been rewritten in the Discussion to:
“Although these symptoms are sometimes blamed on poor compliance or dietary mistakes, they are more often rooted in altered post-surgical physiology. In our patient, recognition of this mechanism through CGM feedback allowed her to identify and correct dietary triggers, which improved symptoms.”
This wording avoids contradiction and better reflects the case findings.

Reviewer 3 Report
Comments and Suggestions for Authors
1) In the title should be reworded to read the statement “case report”
2) Avoid the adjectives to describe diseases in the title as well as the entire manuscript, such as diabetic, obese etc., since they are highly stigmatizing
3) The affiliations of authors are highly stigmatizing
4) The short title should be removed since it is not required according to MDPI guidelines
5) The keywords should differs from those who appear in the title as well in the abstract, otherwise repeated terms become redundant
6) The introduction section is short and poor should be enriched by adding at least a paragraph or two, as well as related references to the topic. Foremost authors should explain what is exceptional in this case to be described as a case report, with respect to the norms
7) Authors before the case presentation should mention that they adhered the CARE Case Report guidelines, which represent the methodology of the paper, hence should be added as a section immediately after the introduction. Moreover a CARE Checklist should be submitted for revision purpose, please indicate each item where is allocated in your manuscript as page and line, please refer to the following link: https://www.care-statement.org/checklist
8) The discussion section should be rewritten as follows:
· The main finding of the study, and its comparison with the available literature published on the topic, and if it in accordance or discordance
· The clinical implications of this case report, what should clinician do or not to do in similar patients
· The strengths and limitations foremost of being a case report and it lack of external validity
· The new directions of the needed future research according to the current finding
9) The reference section is poor (only 9 references) and need to be increases since we are in front of a very common disease (i.e. obesity, type 2 diabetes, bariatric surgery etc.)
Author Response
Comment 1:
“In the title should be reworded to read the statement ‘case report’.”
Response:
We thank the reviewer for this suggestion. The title has been revised to explicitly include “a case report” in line with CARE guidelines and journal indexing standards.
- Previous title: Continuous Glucose Monitoring improves weight loss and hypoglycemic symptoms in a non-diabetic bariatric patient 14 years after RYGB
- Revised title: Continuous glucose monitoring to support self-management in a person without diabetes 14 years after Roux-en-Y gastric bypass: a case report
This updated title avoids stigmatizing adjectives, tones down causal language, and complies with case report formatting requirements.
Comment 2:
“Avoid the adjectives to describe diseases in the title as well as the entire manuscript, such as diabetic, obese etc., since they are highly stigmatizing.”
Response:
We fully agree and have revised the manuscript to eliminate stigmatizing adjectives. Person-first, neutral terminology is now used consistently throughout:
- Title: Changed from “…in a non-diabetic bariatric patient…” to
“…in a person without diabetes 14 years after Roux-en-Y gastric bypass: a case report.” - Abstract and main text:
- “non-diabetic patient” → “person without diabetes”
- “diabetic patients” → “people with diabetes”
- “obese patient” → “individual with obesity”
- “bariatric patient” → “person who has undergone bariatric surgery”
We carefully reviewed the entire manuscript to ensure these changes were applied consistently, aligning the text with current best practices and reducing stigmatizing language.
Comment 3:
“The affiliations of authors are highly stigmatizing.”
Response:
We appreciate the reviewer’s concern. However, author affiliations must be reported exactly as the official names of the respective departments and institutions. These designations are required for accurate attribution, indexing, and institutional recognition, and therefore cannot be altered by the authors.
We have, however, ensured that the rest of the manuscript adheres to person-first, non-stigmatizing language throughout (e.g., replacing “diabetic patient” with “person with diabetes”), in line with current best practice.
Comment 4:
“The short title should be removed since it is not required according to MDPI guidelines.”
Response:
We thank the reviewer for pointing this out. The short title has been removed in accordance with MDPI guidelines.
Comment 5:
“The keywords should differ from those who appear in the title as well as the abstract, otherwise repeated terms become redundant.”
Response:
We thank the reviewer for this helpful suggestion. In the revised manuscript, we replaced overlapping keywords with alternative, non-redundant terms that broaden indexing scope while maintaining relevance. The final keyword list is:
- Post-bariatric hypoglycaemia
- Dumping syndrome
- Glucose variability
- Continuous glucose monitoring
- Patient empowerment
This set avoids duplication with the title and the abstract, while improving search visibility.
Comment 6:
“The introduction section is short and poor… should be enriched by adding at least a paragraph or two, as well as related references. Foremost, authors should explain what is exceptional in this case to be described as a case report, with respect to the norms.”
Response:
We thank the reviewer for this important observation. The Introduction has been substantially revised and expanded:
- Additional background: We added two new paragraphs summarizing the effects of Roux-en-Y gastric bypass (RYGB) on glucose metabolism, the prevalence and pathophysiology of post-bariatric hypoglycaemia (PBH), and the role of continuous glucose monitoring (CGM) in this population.
- Recent literature: We incorporated multiple recent PubMed references to strengthen the background (e.g., Kefurt et al. Surg Obes Relat Dis 2015; Lupoli et al. Nutr Metab Cardiovasc Dis 2020; Yang et al. Obes Surg 2023).
- Exceptional aspect highlighted: We now explicitly explain why this case warrants reporting: it describes a person without diabetes, more than a decade post-RYGB, who used CGM feedback to independently identify and correct dietary triggers, achieving symptomatic improvement and sustained weight loss without formal dietary counseling. Such use of CGM as a facilitator of self-management in long-term post-bariatric follow-up has rarely been documented.
We believe these revisions strengthen the Introduction, provide a richer context for the reader, and clarify the novelty of the present case report.
Comment 7:
“Authors before the case presentation should mention that they adhered to the CARE Case Report guidelines… this should be added as a section immediately after the introduction. Moreover, a CARE Checklist should be submitted for revision, indicating where each item is located (page and line).”
Response:
We thank the reviewer for this important recommendation. In the revised manuscript we have:
- New section added: Immediately after the Introduction, we included a short Methodology section stating that the report was prepared in accordance with the CARE (CAse REport) guidelines (PMID: 24155002).
- Checklist prepared: A completed CARE Checklist has been prepared and submitted as a supplementary file. For each checklist item, we have indicated the corresponding page and line numbers in the manuscript where it is addressed.
- Clarity ensured: This addition explicitly establishes adherence to reporting standards and improves transparency.
Comment 8:
“The discussion section should be rewritten as follows:
· The main finding of the study, and its comparison with the available literature published on the topic, and if it in accordance or discordance
· The clinical implications of this case report, what should clinician do or not to do in similar patients
· The strengths and limitations foremost of being a case report and its lack of external validity
· The new directions of the needed future research according to the current finding.”
Response:
We thank the reviewer for this structured guidance. In the revised manuscript, the Discussion has been fully reorganized according to these four points:
- Main finding and comparison with literature:
- We emphasize the central observation that CGM facilitated dietary self-management and symptom relief in a person without diabetes, more than a decade after RYGB.
- We compare our findings with published studies (e.g., Kefurt et al., Lupoli et al., Ramos-Levi et al., Quevedo et al.) showing that CGM detects more hypoglycaemic episodes than mixed-meal tests and can uncover nocturnal/asymptomatic events.
- We note that our findings are in accordance with this literature, though unique in documenting sustained, patient-led improvement without formal dietary counseling.
- Clinical implications:
- We highlight that clinicians should not dismiss nonspecific post-bariatric symptoms as poor compliance alone, but should consider glucose variability and PBH.
- We stress that CGM should not be used as a stand-alone diagnostic test for PBH, but can be considered as a short-term monitoring/education tool to support dietary adaptation in selected patients.
- Strengths and limitations:
- We present the strength of this report as a clear, well-documented example of CGM-enabled self-management in long-term follow-up.
- We note limitations: the inherent lack of external validity as a single case report, the short 14-day monitoring period, and the confounding effect of dietary changes.
- Future research:
- We recommend larger prospective studies to clarify the role of CGM in post-bariatric follow-up, especially comparative trials of CGM-guided dietary self-management versus standard nutritional counseling.
- We also suggest investigating integration of CGM into multidisciplinary bariatric follow-up to improve quality of life and detect PBH earlier.
We believe the revised Discussion now fully addresses the reviewer’s concerns and provides a balanced, structured interpretation of the case.
Comment 9:
“The reference section is poor (only 9 references) and needs to be increased since we are in front of a very common disease (i.e. obesity, type 2 diabetes, bariatric surgery etc.).”
Response:
We thank the reviewer for this observation. In the revised manuscript, we substantially enriched the reference list to better reflect the common and well-studied nature of obesity, type 2 diabetes, bariatric surgery, and post-bariatric hypoglycaemia.
- New references added: We expanded from 9 references to 18, incorporating recent PubMed-indexed studies and meta-analyses on:
- The global burden of obesity and type 2 diabetes (Goday et al. 2021; Jaques-Albuquerque et al. 2023).
- Trends in bariatric surgery worldwide and in the U.S. (Lazzati et al. 2023; Altieri et al. 2021).
- CGM use in post-bariatric populations, including observational studies, meta-analyses, and reviews (Lupoli et al. 2020 & 2022; Kefurt et al. 2015; Turk et al. 2025; Quevedo et al. 2024; Karimi et al. 2024; Bjerkan et al. 2024).
- Integration in text: These references are now cited in the Introduction to provide epidemiologic context, and in the Discussion to support comparison with existing literature and highlight the novelty of our case.
This expansion strengthens the scholarly foundation of the manuscript and places our case report in the broader context of current evidence.

Round 2
Reviewer 2 Report
Comments and Suggestions for Authors
I have carefully reviewed the revised version of the manuscript and I am satisfied with the changes made. I recommend the article for publication.
Reviewer 3 Report
Comments and Suggestions for Authors.